# Novel *PKD2* Missense Mutation p.Ile424Ser in an Individual with Multiple Hepatic Cysts: A Case Report

**DOI:** 10.3390/medicines9040025

**Published:** 2022-03-29

**Authors:** Seiko Miura, Yo Niida, Chieko Hashizume, Ai Fujii, Yuta Takagaki, Kahoru Kusama, Sumiyo Akazawa, Tetsuya Minami, Tsuyoshi Mukai, Kengo Furuichi, Mutsumi Tsuchishima, Nobuhiko Ueda, Hiroyuki Takamura, Daisuke Koya, Tohru Ito

**Affiliations:** 1Department of General and Gastrointestinal Surgery, Kanazawa Medical University, 1-1 Daigaku, Uchinada, Kahoku 920-0293, Ishikawa, Japan; ueda215@kanazawa-med.ac.jp (N.U.); takamuh@kanazawa-med.ac.jp (H.T.); 2Women’s Health Center, the Department of General Medicine, Kanazawa Medical University, 1-1 Daigaku, Uchinada, Kahoku 920-0293, Ishikawa, Japan; 69kahorucl@gmail.com (K.K.); sumiyok@kanazawa-med.ac.jp (S.A.); mutsumi@kanazawa-med.ac.jp (M.T.); 3Department of Advanced Medicine, Kanazawa Medical University, 1-1 Daigaku, Uchinada, Kahoku 920-0293, Ishikawa, Japan; niida@kanazawa-med.ac.jp; 4Department of Hepatology, Kanazawa Medical University, 1-1 Daigaku, Uchinada, Kahoku 920-0293, Ishikawa, Japan; hashi@kanazawa-med.ac.jp; 5Department of Nephrology, Kanazawa Medical University, 1-1 Daigaku, Uchinada, Kahoku 920-0293, Ishikawa, Japan; fujiiai@kanazawa-med.ac.jp (A.F.); furuichi@kanazawa-med.ac.jp (K.F.); 6Department of Diabetology and Endocrinology, Kanazawa Medical University, 1-1 Daigaku, Uchinada, Kahoku 920-0293, Ishikawa, Japan; takagaki@kanazawa-med.ac.jp (Y.T.); koya0516@kanazawa-med.ac.jp (D.K.); 7Department of Radiology, Kanazawa Medical University, 1-1 Daigaku, Uchinada, Kahoku 920-0293, Ishikawa, Japan; tminami@kanazawa-med.ac.jp; 8Department of Gastroenterological Endoscopy, Kanazawa Medical University, 1-1 Daigaku, Uchinada, Kahoku 920-0293, Ishikawa, Japan; tsmukai@kanazawa-med.ac.jp (T.M.); itotohru@kanazawa-med.ac.jp (T.I.)

**Keywords:** autosomal dominant polycystic kidney disease (ADPKD), *PKD2* (polycystic kidney disease type 2), polycystic liver, estrogen

## Abstract

We report a novel missense mutation, p.Ile424Ser, in the *PKD2* gene of an autosomal dominant polycystic kidney disease (ADPKD) patient with multiple liver cysts. A 57-year-old woman presented to our university hospital with abdominal fullness, decreasing appetite, and dyspnea for three months. A percutaneous drainage of hepatic cysts was performed with no significant symptomatic relief. A computed tomography (CT) scan revealed a hepatic cyst in the lateral portion of the liver with appreciable compression of the stomach. Prior to this admission, the patient had undergone three drainage procedures with serial CT-based follow-up of the cysts over the past 37 years. With a presumptive diagnosis of extrarenal manifestation of ADPKD, we performed both a hepatic cystectomy and a hepatectomy. Because the patient reported a family history of hepatic cysts, we conducted a postoperative genetic analysis. A novel missense mutation, p.Ile424Ser, was detected in the *PKD2* gene. Mutations in either the *PKD1* or *PKD2* genes account for most cases of ADPKD. To the extent of our knowledge, this point mutation has not been reported in the general population. Our in-silico analysis suggests a hereditary likely pathogenic mutation.

## 1. Introduction

Studies have shown that mutations in one of two genes, *PKD1* (polycystin 1) at position 16p13.3 and *PKD2* (polycystin 2) at 4q22.1, account for 80% and 15% of ADPKD cases, respectively [1]. Although there has been increasing evidence regarding the molecular pathogenesis of renal cyst formation, the mechanism of hepatic cyst formation has not been elucidated.

We report a novel missense mutation, p.Ile424Ser, in the *PKD2* gene of an autosomal dominant polycystic kidney disease (ADPKD) patient with multiple liver cysts. To the extent of our knowledge, this point mutation has not been reported in the general population. Our in-silico analysis suggests a hereditary likely pathogenic mutation.

## 2. Case Presentation

A 57-year-old female, with an established history of multiple liver cysts, presented to the Internal Medicine Department of our university hospital with a three-month history of epigastric discomfort and reduced appetite. The patient’s hepatic cysts were incidentally found on a computed tomography (CT) scan at 20 years of age. There existed a family history of bilateral renal cysts and multiple liver cysts in her maternal grandfather and mother and liver cancer in her father. The patient previously underwent three rounds of percutaneous drainage of hepatic cysts at ages 48, 53, and 54 years. A percutaneous cyst drainage, CT-guided, was performed this time as well, by the treating physician at the Interventional Radiology Department in collaboration with the Internal Medicine Department; however, it led to minimal symptomatic relief. Subsequently, the patient was referred to our surgical department for further evaluation and treatment of the hepatic cysts.

A physical examination showed mild icterus and palpable liver with associated mild abdominal tenderness. The patient was found to be mildly anemic with a hemoglobin level of 10.5 g/dL. Moreover, she presented with hypokalemia with a potassium level of 2.5 mEq/L and elevated levels of biliary enzymes, including γGTP (117 U/L) and alkaline phosphatase (564 U/L). The level of follicle-stimulating hormone was 83.3 mIU/mL, and that of estradiol was 8 pg/mL. The alpha-fetoprotein level was slightly elevated to 16.5 ng/mL (≤10.0). The indocyanine green plasma disappearance rate (K value) was 0.202, which was within the normal range. The level of serum creatine was 0.8 mg/dL, eGFR 80 mL/min/1.73 m^2^; urinalysis was not particular; and systolic blood pressure was 110 mmHg.

Abdominal magnetic resonance imaging (MRI) T2 horizontal section and magnetic resonance cholangiopancreatography revealed no dilation of the intrahepatic bile duct, but cystic lesions of various sizes were scattered across both the lobes. The cysts protruding from the right lobe, S5, were slightly heterogenous and septate with low signal intensity, indicating intracystic bleeding. Some cysts showed a decrease in size after drainage, but the extrahepatic cysts exhibited no change. These extrahepatic laterally located cysts measured up to 90 mm in diameter and were found to be compressing the stomach, causing a posterior deviation. In addition, numerous bilateral renal cysts were visualized (Figure 1). Thus, we presumptively diagnosed the patient as having ADPKD with multiple liver cysts and performed a hepatectomy for symptomatic relief.

Intraoperative findings were found to be consistent with the MRI findings. A cyst protruding from the left lobe of the liver compressed the stomach, causing a posterior deviation. Hence, a left hepatic lobectomy, a caudate lobectomy, an S5 cystectomy, and a cholecystectomy were performed. Intraoperative cytological analysis was negative for malignancy. In addition, we performed fenestration of the cysts located on the liver surface and incineration of the luminal surface (Figure 2).

The patient suffered no complications during the one-year postoperative follow-up period. No change in cyst size or extent with CT was noted upon initiation of nonhormonal management of perimenopausal symptoms.

Lastly, genetic analysis was performed, and searches in dbSNP; https://www.ncbi.nlm.nih.gov/snp/ (accessed on 13 March 2022) and genomAD; https://gnomad.broadinstitute.org/ (accessed on 13 March 2022), revealing a novel missense variant NM_000297.4:c.1271T>G p.Ile424Ser in the *PKD2* gene (Figure 3). This missense variant is located in the polycystin domain of the PKD2 protein [2] and is a rare variant that has not been reported in the general population on the database. Multiple lines of computational evidence support a deleterious effect (Table 1), and the patient’s phenotype and family history were consistent with ADPKD. According to the guideline of the American College of Medical Genetics and Genomics [3], ADPKD is classified as likely pathogenic. We used a very long amplicon sequence method [4] to screen for mutations in the *PKD1* and the *PKD2*, but no other mutations were detected. Further, all long PCR amplicons maintained heterozygosity and no large deletions/insertions were detected.

## 3. Discussion

ADPKD was reported by Bear et al. in 1984, Ravine et al. in 1994, and Pei et al. in 2009, and evidence-based guidelines for the treatment of polycystic kidney disease were published in Japan in 2017 [5,6,7,8]. The diagnosis of ADPKD is based on the following criteria: confirmed familial disease; onset at 16 years of age or older; presence of three or more cysts on abdominal CT, MRI, or ultrasonography; and exclusion of diseases, including simple renal cysts, tubular acidosis, multicystic kidney, multifocal cysts of the kidney, medullary cystic disease of the kidney, acquired cystic disease of the kidney, and autosomal recessive polycystic kidney disease [8]. In our case, there were more than five bilateral renal cysts, with no renal dysfunction in the course of the disease and no extrarenal lesions other than hepatic cysts, and the identified PKD2 mutation was suspected to be the main cause of the multiple hepatic cysts. Although previous studies have mentioned that the majority of ADPKD cases are caused by a mutation in the *PKD1* gene, they failed to mention that in the case of a *PKD1* mutation, the presentation of the disease is earlier, and the disease is more likely to progress to end stage renal failure (ESRF), while the presentation in the case of a *PKD2* mutation is less severe and less likely to lead to ESRF. The most common extrarenal lesion in ADPKD occurs in the form of polycystic liver disease, which is classified via Gigot classification based on the number, size, and distribution of cysts observed on CT to evaluate the severity of the disease. Accordingly, sclerotherapy, coronary embolization, cyst opening, hepatectomy, or liver transplantation is performed to treat the lesions. Symptoms of multiple hepatic cysts are mainly compression symptoms, including gastrointestinal disturbances such as abdominal distention, upper abdominal pain, and decreased food intake; respiratory disturbances due to compression of the lungs and heart, pain, and changes in vital signs because of rupture; and anemia due to bleeding. In addition, if there appears to be an irregular thickening of the cyst wall or a contrast effect upon imaging, mucinous cystic neoplasm or intraductal papillary neoplasm of the bile duct cannot be ruled out, and malignant transformation or precancerous lesions may be detected. Therefore, the patient is subject to surgical resection to eliminate malignant transformation and precancerous lesions. 

In our case, the patient was classified as Gigot type II (diffusely distributed small-to-medium-sized scales in the liver, along with the existence of some degree of normal liver parenchyma without scales), and surgical resection was effective [9]. Since her maternal grandfather and mother also showed a history of polycystic kidney disease and multiple hepatic cysts, the patient was apprised of the possibility of ADPKD, and a genetic diagnosis was made after genetic counseling. We found a novel missense variant p.Ile424Ser in the *PKD2* gene, which was thought to be one of the causative genes of the disease. This missense variant has not been registered in the general population previously, and our in-silico analysis strongly suggested that it was a pathogenic mutation.

To date, ADPKD has been reportedly caused by a decrease in the intracellular levels of polycystin proteins PC1 (membrane protein) and PC2 (cationic transient receptor potential channel), encoded by the *PKD1* (chromosomal locus 16p13.3) and *PKD2* genes (chromosomal locus 4q22.1), respectively, which form heterodimers in the cilia of cells (Figure 4). A reduced heterodimer formation in the cilia of cells leads to cyst formation, and the molecular mechanism of renal cyst formation has been clarified [10,11,12,13,14,15]. Additionally, the variant p.Ile424Ser is in β3 sheet of PC2 [2]. The protein structure analysis revealed that a groove formed between finger 1 and the β4-β5 turn of one polycystin domain interacts with the β3-β4 turn and H1 helix of a neighboring polycystin domain. Therefore, we considered that the variant p.Ile424Ser may alter the β3 sheet structure, preventing it from interacting with other domains, and then prevent channel assembly and suppress the function.

However, mechanisms underlying the formation of multiple hepatic cysts remain to be clarified. These mechanisms may be specific to hepatic cysts derived from the bile duct epithelium. Hence, as the gene mutation has been identified in the present study, future research should aim at providing further evidence for the pathogenicity of this mutation and revealing possible mechanisms of hepatic cysts development. In addition to the findings in regular imaging analysis, the frequency of occurrence of multiple hepatic cysts in patients with ADPKD ranges from 20% (for patients in their twenties) to 75% (for patients in their fifties) [16]. Multiple hepatic cysts occur earlier in women than in men and are more severe in women who have had multiple childbirths than in other women. It has been speculated that estrogen plays an important role in the development of hepatic cysts, as cyst enlargement is seen in women receiving estrogen replacement therapy after menopause [17,18,19]. In our patient, who had a history of three childbirths, estrogen might have been involved in the enlargement of the cysts, raising a concern that the cysts might continue to enlarge. At present, a year has passed since the surgery, and there are no signs of cyst enlargement on imaging.

## 4. Conclusions

We found a novel *PKD2* missense mutation p.Ile424Ser in a patient with ADPKD mainly presenting with polycystic liver disease, and our in-silico analysis strongly indicated that this mutation is a likely pathogenic mutation. Hepatic cystectomy and hepatectomy were performed to control the disease. We believe our findings indicate the pathogenic nature of the *PKD2* mutation and provide a clue to the molecular mechanism of hepatic cyst formation.

## Figures and Tables

**Figure 1 medicines-09-00025-f001:**
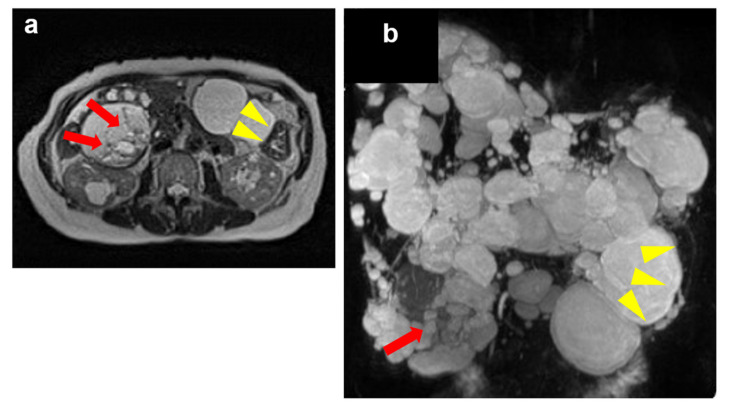
Magnetic resonance imaging of the polycystic liver performed during the patient’s first visit. (**a**) Axial T2-weighted image. (**b**) Magnetic resonance cholangiopancreatography revealed intracystic heterogeneity and septation with associated focal hyperintensity, suggestive of intracystic bleeding (red arrows). Extrahepatic cysts compress the stomach (yellow arrowheads).

**Figure 2 medicines-09-00025-f002:**
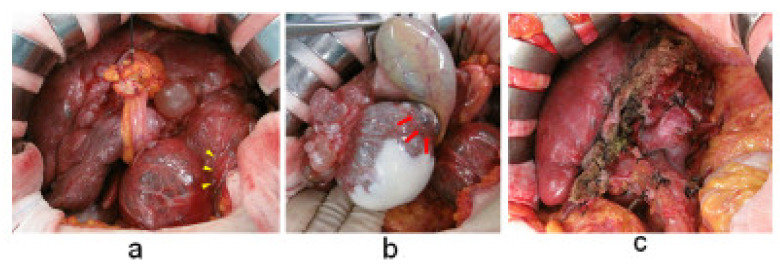
Surgical findings. (**a**) Multiple hepatic cysts occupy the upper abdominal cavity. Extrahepatic cysts compress the stomach (yellow arrowheads). (**b**) Intracystic bleeding scars are present. Intraoperative cytological analysis of the cyst content was negative for malignancy (red arrows). (**c**) Abdominal cavity after cystectomy and hepatectomy.

**Figure 3 medicines-09-00025-f003:**
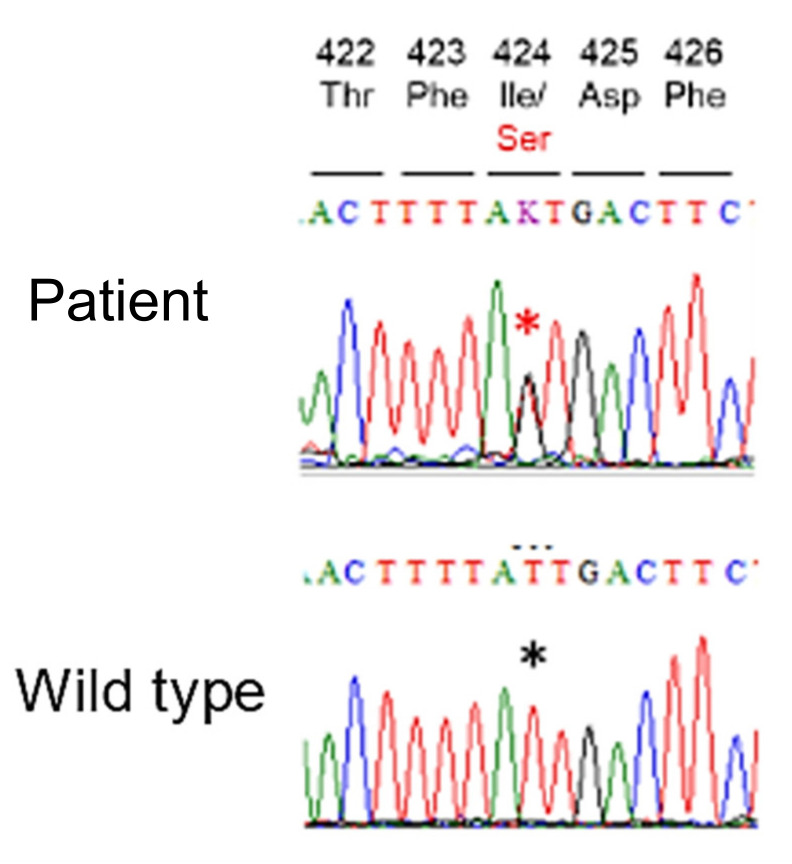
Genetic analysis reveals a novel missense mutation in the *PKD2* gene of the patient. DNA sequence codes: A (adenine), C (cytosine), G (guanine), T (thymine) and K (T or G). Red asterisk indicates T to G mutation. Black asterisk is wild-type gene sequence, on the same position as red asterisk. The DNA sequence of the patient has one allele wild type T and the other has a mutation G. The upper numbers 422 to 426 indicate aa (amino acid) residue of PC2.

**Figure 4 medicines-09-00025-f004:**
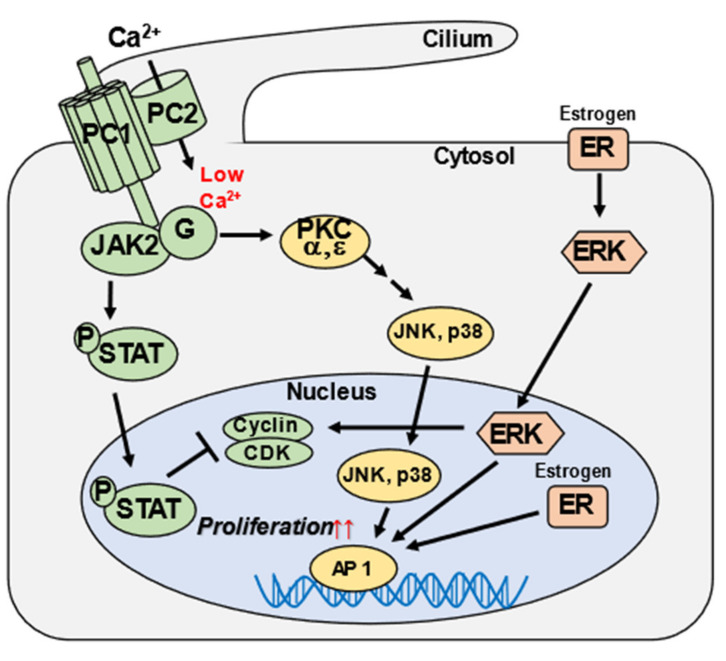
Signaling pathways involved in autosomal dominant polycystic kidney disease (ADPKD) pathogenesis. ADPKD is caused by mutations in *PKD1* and *PKD2*, which code for the polycystin proteins PC1 and PC2, respectively, which form heterodimers in the cilia of cells. PC1 and PC2 have been reported to function in various signaling pathways. PC1 and PC2 bind and form part of a protein complex found in the primary cilium that operates as a flow-regulated Ca^2+^ channel. Defects in any of the complex proteins reduce Ca^2+^ influx and lead to low Ca^2+^ intracellular concentrations. The PC1/PC2 complex interacts with and activates JAK2, which phosphorylates STAT. STAT normally inhibits the cyclin/CDK pathways, which promote cell cycle progression; hence, its downregulation in ADPKD would increase epithelial cell proliferation. PC1 and PC2 signal via G proteins and distinct PKC isoforms to activate JNK and p38, which in turn stimulate transcription factor AP1. Estrogen, acting through the cell membrane and cytosolic/nuclear receptors (ER), can also stimulate cell proliferation. This figure is adapted from Figure 1 in a previous study, “Somatostatin, estrogen, and polycystic liver disease” [19]. Modified and reprinted from ref [19] Copyright (2013), with permission from Elsevier [License Number: 5275740748526].

**Table 1 medicines-09-00025-t001:** Summary of in-silico analysis.

Gene Analysis Soft Ware	SIFT	PolyPhe-2 HumDiv	PolyPhe-2 HumVar	Mutation Taster	Mutation Assessor	Likelihood Ratio Test (LRT)	CADD PHRED Like Scaled C-Score (>20)
result	Damaging (0.000)	Possibly damaging (0.897)	Probably damaging (0.990)	Disease Causing (0.9998)	Medium (3.02)	Deleterious (0.000209)	29.8

SIFT (D = damaging, T = tolerated), PolyPhen-2 with HumDiv training set (D = probably damaging, P = possibly damaging, B = benign), PolyPhen-2 with HumVar training set (D = probably damaging, P = possibly damaging, B = benign), MutationTaster (A = disease causing automatic, D = disease causing, N = polymorphism, P = polymorphism automatic), MutationAssessor (high or medium: predicted functional impact; low or neutral: predicted non-functional impact), Likelihood ratio test (LRT) (D = deleterious, N = Neutral, U = unknown). SIFT and LRT are ranked on a scale of 0 to 1, with closer to 0 being more morbid. PolyPhen2, Mut Taster is ranked from 0 to 1, the closer to 1, the more morbid. CADD has a cutoff of >20 as pathological.

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
