# Peer review of "Novel PKD2 Missense Mutation p.Ile424Ser in an Individual with Multiple Hepatic Cysts: A Case Report"

_medicines, 2022, doi:10.3390/medicines9040025_

Round 1

Reviewer 1 Report

Line 50-54: The authors mention that the patient underwent several rounds of percutaneous drainage of hepatic cysts prior to surgery without relief of symtpoms. It would be beneficial to add which exact technique was used to drain the cysts (CT/US or MRI guided, was additional scleropherapy or other procedure perfomed) and whether there was any follow-up. For example, it is not clear were the symtoms caused by a newly developed cyst or a reexpanding cysts that was previously drained.

Line 64-67: The MRI protocol used for imaging should be added. Line 66: It should be stated on which MRI sequences were the cysts heterogenous (some of the info is stated in the description of Figure 1 and can be added to the text as well).

Line 79-81: The authors should add how the follow-up was perfomed. Was imaging perfomed to comfirm no change in cyst size (mentioned in line 150)? If yes, which modality and how often?

Line 100-103: The authors mentioned that there was no renal dysfunction in the course of the disease for this patient. Altought they have mentioned that the majority of ADPKD cases are caused by a mutation in the PKD1 gene they failed to mention than in the case of a PKD1 mutation the presentation of the disease is earlier and the disease is more likely to progress to end stage renal failure (ESRF) while the presentation in the case of a PKD2 mutation is less severe and less likey to lead to ESRF (as it is in this case). It would be beneficial to add this into the text since it provides an important clinical correlate and explans why the patient had normal renal function despite lifelong disease.

Reviewer 2 Report

This article reports a novel missense mutation in the PKD2 gene in a patient with ADPKD and polycystic liver disease. The authors provide some preliminary evidence for the pathogenicity of this mutation. However, its clinical significance is still unclear.

Major comments:

1. Please indicate the specific criteria met in the guideline for interpreting the mutation as “likely pathogenic”. It does not appear that there is sufficient evidence to support such assertion as only one piece of moderate evidence and two pieces of supporting evidence are available.

2. The authors claim that this point mutation has not been reported in the general population. Please indicate which databases were searched.

3. I recommend the authors provide more information about this missense mutation and its impact on the structure and function of the gene product. What is the location and context within the protein sequence (which exon, transmembrane domains or cytoplasmic N and C termini)? Does the variant occur in a conserved nucleotide? Which is the biochemical consequence of the amino acid substitution? 

4. Has genetic testing been conducted for the patient’s maternal grandfather and mother? Do they have the same missense mutation?

5. The authors claim that the mutation is de novo. However, the patient clearly has a family history.

6. Important positive and negative clinical findings of ADPKD should be mentioned in the case presentation (e.g., serum creatine level and eGFR, urinalysis, blood pressure).

7. In my opinion, the discussion should focus more on the genetics of ADPKD and the significance of the new-discovered mutation. The discussion about the clinical issues can be shortened.

Minor comments:

It should be “p.Ile424Ser”. The letter “L” is miswritten as the number “1” throughout the article.

Line 26-27: The sentence “No change in either cyst size or extent was seen after initiating a non-hormonal management of perimenopausal symptoms” is unimportant and could be deleted from the abstract.

Line 95-100: Please provide references for such diagnostic criteria. The diagnosis of ADPKD does not necessarily require positive family history or age of onset after 16.

Line 118-123: This information has already been provided in the case presentation. The paragraph can be shortened.

Line 127: “…, which was thought to be one of the causative genes of the disease” would be more accurate.

Line 138-140: “Hence, …, future research should likely be conducted on…” The rationale for this sentence is not clear. In my opinion, future research should aim at providing further evidence for the pathogenicity of this mutation and revealing possible mechanisms of hepatic cysts development.

Line 153-154: The conclusion should be in line with the guidelines for interpreting sequence variants. I think this mutation is “likely pathogenic” at most based on current evidence. The authors should interpret their findings with caution.

Line 155-156: “We believe our findings would pave the way for clarifying the role of PKD2 mutations in hepatic cyst formation” I think this statement is exaggerated.

Figure 1, Line 161: “Axial T2-weighted image” would be more appropriate.

Figure 2a: It is unclear what the yellow arrowheads are pointing to.

Figure 3: Please indicate what the letter K means.

Table 1: Please indicate the meaning of the numbers in the table.
